# TheFinite Element Modeling and Experimental Study of Sandwich Plates with Frequency-Dependent Viscoelastic Material Model

**DOI:** 10.3390/ma13102296

**Published:** 2020-05-15

**Authors:** Zhicheng Huang, Xingguo Wang, Nanxing Wu, Fulei Chu, Jing Luo

**Affiliations:** 1Jingdezhen Ceramic Institute, College of Mechanical and Electrical Engineering, Jingdezhen 333001, China; huangzhicheng@jci.edu.cn (Z.H.); wunanxing@jci.edu.cn (N.W.); 2Department of Mechanical Engineering, Tsinghua University, Beijing 100084, China; chufl@mail.tsinghua.edu.cn; 3Beijing Research Institute of Automation for Machinery Industry Co., Ltd., Beijing 100120, China; sactc3@riamb.ac.cn

**Keywords:** viscoelastic sandwich plate, viscoelastic material, Biot model, finite element method, vibration characteristics

## Abstract

Athree-layer composite plate element is developed for finite element modeling and vibration analysis of sandwich plate with frequency-dependent viscoelastic material core. The plate element is quadrilateral element bounded by four-node with 7-degree-of-freedom per node. The frequency-dependent characteristics of viscoelastic material parameters are described using the Biot model. The method of identifying the parameters of the Biot model is given. By introducing auxiliary coordinates, the Biot model is combined with the finite element equation of the viscoelastic sandwich plate. Through a series of mathematical transformations, the equation is transformed into a standard second-order steady linear system equation form to simplify the solution process. Finally, the vibration characteristics of the viscoelastic sandwich plate are analyzed and experimentally studied. The results show that the method in this paper is correct and reliable, and it has certain reference and application value for solving similar engineering vibration problems.

## 1. Introduction

Many materials have viscoelastic properties [1,2,3], among which viscoelastic materials have excellent energy dissipation properties [4]. In engineering, they are usually laminated with elastic materials to construct viscoelastic sandwich structures, which are widely used for vibration suppression of the thin-walled structures in aerospace, automotive and ship equipment [5,6,7,8]. Figure 1 shows a viscoelastic sandwich plate structure. A viscoelastic material layer with a high loss factor is sandwiched by the base plate and the constraining layer. When the base plate is subjected to vibration, the viscoelastic layer will undergo shear deformation, which converts vibration energy into heat and dissipates it. This structure can effectively suppress the vibration without significantly changing the weight of the components. Therefore, it is widely used in the situation of strict weight restrictions [9]. For decades, the studies on the dynamic modeling, vibration and damping characteristics of the viscoelastic sandwich plates have been hot topics.

The finite element method is commonly used in engineering applications to study the vibration of viscoelastic sandwich plate structures. Johnson et al. [10] used three-dimensional solid elements to model three-layer viscoelastic composite structures in the commercial finite element software NASTRAN. The viscoelastic layer was divided by solid element (HEXA8), and the elastic surface layer was divided by quadrilateral thick shell element (QUAD4). Plouin et al. [11] also used a similar method, in which the elastic surface layer and the viscoelastic core layer were modeled by the traditional shell element and the solid element, respectively. These methods are complex and time-consuming. In order to improve these problems, some composite elements were used to model sandwich plate structures. Chen et al. [12] proposed four types of three-layer composite elements to study the dynamic characteristics of viscoelastic composite plate structures. Park et al. [13] derived two kinds of finite element models of active constrained layer damping plates based on classical and laminated plate theory, respectively, and compared their accuracy and efficiency. Huang et al. [14,15] studied the vibration characteristics of sandwich plate using three-layer composite plate elements. Some other finite element methods were also used to study the vibration of sandwich plates. Zhao et al. [16] proposed the finite element model for the sandwich plate by using a single-layer equivalent method, which was based on the equivalent material properties. Zhang et al. [17] established a sound radiation optimization model for sandwich plates. The finite element method was used to formulate the normal vibration velocities, and the boundary element method was used to derive the sound power. Kumar et al. [18] developed a finite element model to investigate the damping mechanisms of the plate structures with 0-3 viscoelastic layer. Ojha et al. [19] carried out a dynamic analysis of sandwich plates with a viscoelastic core using finite element method.

The parameters of viscoelastic materials are frequency-dependent, which leads to the difficulty of mathematical modeling of viscoelastic sandwich structures. Most of the above works did not consider the frequency dependence of viscoelastic materials. In order to solve this problem, some viscoelastic material constitutive models were used to consider the frequency-dependent characteristic. Ayodele et al. [20] presented a finite element model for sandwich structures with multi-layered, frequency-dependent viscoelastic cores based on the zig-zag approach. Hamdaoui et al. [21] compared the computational times and accuracy of the non-linear Arnoldi method, non-linear Jacobi–Davidson method, inverse iteration and iterative shift-invert method on relevant use cases with frequency dependent visco-elastic materials. Alvelid and Enelund developed an interface finite element for sandwich structures. The constitutive behavior of the viscoelastic material was described by a fractional order viscoelastic model [22].Hamdaoui et al. [23] used an adjoint method to identify the viscoelastic parameter of frequency-dependent viscoelastic damped structures. Wang et al. [24] investigated the frequency-dependent viscoelastic dynamics of viscoelastic composite structure by finite element analysis and experimental validation. The GHM and ADF approaches are used to implement the viscoelastic material model. Rijnen et al. [25] studied the viscoelastic damping of a 3D structure. The fractional derivative model was used to simulate the viscoelastic materials. Xie et al. [26] proposed a strategy for the vibration analysis of viscoelastic damping structures. Three damping models, called the GHM model, generalized Maxwell model and fractional derivative model were used to describe the frequency dependency of the viscoelastic materials. Kumar et al. [27] introduced the frequency-dependent Young’s modulus and loss factors in power series in the sandwich plate finite model by using an iterative scheme. Huang et al. [28] studied the damping mechanism of viscoelastic sandwich structures by modeling viscoelastic materials with a GHM model. In the above models, the GHM leads to too many dimensions of system equations; the generalized Maxwell model needs to obtain the performance parameters of viscoelastic materials in a wide frequency range, which will cause difficulties in practical applications, and the fractional derivative model has a large amount of calculation in the vibration analysis of viscoelastic composite structure.

This work presents a new finite element method combined with Biot model for frequency-dependent viscoelastic sandwich plates. The Biot model is used to consider the frequency-dependent properties of viscoelastic materials, and the method to determine its parameters is presented. A type of three-layer four-node 28-degree-freedom composite plate element is developed for finite element modeling of the viscoelastic sandwich plate structure. By introducing auxiliary coordinates, the Biot model is incorporated into the finite element equation of the viscoelastic sandwich plate, and then transforms it into a standard second-order differential equation form to simplify the solution process. Finally, the vibration characteristics of viscoelastic sandwich plates are numerically analyzed and experimentally researched. The results show that the method presented this paper is correct and reliable.

## 2. Finite Element Modeling for the Sandwich Plate

### 2.1. Assumptions

It is assumed that the plate satisfies the Kirchhoff–Love hypothesis. The shear strains of the two elastic surface layers (the constraining layer and the base plate) are ignored, and only the shear strain of the viscoelastic layer is considered. The deflections of the layers in the thickness direction can be ignored, that is, the three layers have the same deflection. The elastic layers do not dissipate vibration energy, the viscoelastic layer is incompressible material, and the vibration energy is dissipated only through its shear deformation. The viscoelastic layer is a linear viscoelastic material. Each layer is perfectly bonded and there is no relative sliding.

### 2.2. Description of Geometry and Kinematics

The geometric deformation relationship of each layer of a sandwich plate in the XOZ plane is shown in Figure 2. $u_{1x},\;u_{2x}\;$ and $u_{3x}$ are the mid-plane displacements of the base plate, the viscoelastic layer and the constraining layer along the *X* direction, respectively. $h_{1},\;h_{2}\;$and $h_{3}$ are the thickness of the base plate, the viscoelastic layer and the constraining layer, respectively. w and ∂w/∂x denote the deflection and the angle around the *y*-axis of the sandwich plate, respectively. $\psi_{x}^{v}$ and $\gamma_{xz}^{v}$ denote the angle around the *Y*-axis and shear strain in the XOZ plane of the viscoelastic layer, respectively.

The displacements and the shear strains of the viscoelastic layer can be determined from the geometry of the sandwich plate in Figure 2 [15],
(1)$$u_{2x}=\frac{1}{2}\left[\left(u_{1x}+u_{3x}\right)+\left(\frac{h_{3}-h_{1}}{2}\right)\frac{\partial w}{\partial x}\right],u_{2y}=\frac{1}{2}\left[\left(u_{3y}+u_{1y}\right)+\left(\frac{h_{3}-h_{1}}{2}\right)\frac{\partial w}{\partial y}\right]$$
(2)$$\gamma_{xz}^{v}=\frac{1}{h_{2}}\left[\left(u_{3x}-u_{1x}\right)+d\frac{\partial w}{\partial x}\right],\gamma_{yz}^{v}=\frac{1}{h_{2}}\left[\left(u_{3y}-u_{1y}\right)+d\frac{\partial w}{\partial y}\right]$$
where $u_{1y},\;u_{2y}$ and $u_{3y}$ are the mid-plane displacements of the base plate, the viscoelastic layer and the constraining layer along the *Y* direction, respectively, $\gamma_{yz}^{v}$ denotes the shear strain of the viscoelastic layer in the YOZ plane, $d=\left(h_{3}+h_{1}\right)/2+h_{2}$ is the mid-plane distance between the two elastic surface layers.

### 2.3. Degrees of Freedom and Shape Functions

The sandwich plate element developed here is shown in Figure 3. It is a rectangular element with the dimension of 2a×2b. Each node has 7 DOF, which respectively represent the longitudinal displacement $u_{1x}$ (x-direction) and $u_{1y}$ (y-direction) of the base plate layer, the longitudinal displacement $u_{3x}$ (x-direction) and $u_{3y}$ (y-direction) of the constraining layer, the transverse deflection w, and the deflection angles $\theta_{x}$ and $\theta_{y}$ of the sandwich plate element.

Over any element *i* of the sandwich plate, their spatial distributions (interpolation functions) can be given by
(3)$$\begin{array}{l} u_{3}=a_{1}+a_{2}x+a_{3}y+a_{4}xy,\;v_{3}=a_{5}+a_{6}x+a_{7}y+a_{8}xy\\ u_{1}=a_{9}+a_{10}x+a_{11}y+a_{12}xy,\;v_{1}=a_{13}+a_{14}x+a_{15}y+a_{16}xy\\ w=a_{17}+a_{18}x+a_{19}y+a_{20}x^{2}+a_{21}xy+a_{22}y^{2}+a_{23}x^{3}+a_{24}x^{2}y\\ +a_{25}xy^{2}+a_{26}y^{3}+a_{27}x^{3}y+a_{28}xy^{3}\\ \theta_{x}=\frac{\partial w}{\partial y},\;\theta_{y}=-\frac{\partial w}{\partial x}\; \end{array}$$
wherethe constant coefficient $a_{1},\;a_{2},\;\ldots ,a_{28}$ are determined by the 28 node displacement vectors $\Delta^{e}$ of the four element nodes 1,2,3 and 4. The displacement vector of the node is given by
(4)$$\Delta^{e}={\left\{\begin{array}{cccc} \Delta_{1} & \Delta_{2} & \Delta_{3} & \Delta_{4} \end{array}\right\}}^{T}$$
where
(5)$$\Delta_{i}={\left\{\begin{array}{ccccccc} u_{3i} & v_{3i} & u_{1i} & v_{1i} & w_{i} & \theta_{xi} & \theta_{yi} \end{array}\right\}}^{T},\;i=1,2,3,4$$

Therefore, the displacement Δ of any position (*x*, *y*) in the *i*th element can be obtained by interpolation of the element node displacement vector, that is
(6)$$\Delta ={\left[\begin{array}{ccccccc} u_{3} & v_{3} & u_{1} & v_{1} & w & \theta_{x} & \theta_{y} \end{array}\right]}^{T}=N\Delta^{e}$$
where $N={\left[\begin{array}{ccccccc} N_{1} & N_{2} & N_{3} & N_{4} & N_{5} & N_{6} & N_{7} \end{array}\right]}^{T}$ are the spatial interpolating vectors (shape function), corresponding to $u_{3},\;\;v_{3}\;,\;\;u_{1},\;\;v_{1},\;w,\;\theta_{x}$ and $\theta_{y}$.

Substituting the shape function N into Equations (1) and (2), respectively, the longitudinal displacement and shear strain of the viscoelastic layer can be obtained as
(7)$$\begin{array}{cccc} u_{2x}=N_{8}\Delta^{e},\; & u_{2y}=N_{9}\Delta^{e},\; & \gamma_{xz}^{v}=N_{10}\Delta^{e},\; & \gamma_{yz}^{v}=N_{11}\Delta^{e} \end{array}$$
where $N_{8},\;N_{9},\;N_{10}$ and $N_{11}$ are the shape functions corresponding to the longitudinal displacement $u_{2x},\;u_{2y}$ and the shear strain $\gamma_{xz}^{v},\;\gamma_{yz}^{v}$ of the viscoelastic layer, respectively, where
(8)$$N_{8}=\frac{1}{2}\left[\left(N_{1}+N_{3}\right)+\left(\frac{h_{3}-h_{1}}{2}\right)(-N_{7})\right]$$
(9)$$N_{9}=\frac{1}{2}\left[\left(N_{2}+N_{4}\right)+\left(\frac{h_{3}-h_{1}}{2}\right)(N_{6})\right]$$
(10)$$N_{10}=\frac{1}{h_{v}}\left[\left(N_{1}-N_{3}\right)+\left(\frac{h_{1}+h_{3}}{2}+h_{2}\right)(-N_{7})\right]$$
(11)$$N_{11}=\frac{1}{h_{v}}\left[\left(N_{2}-N_{4}\right)+\left(\frac{h_{3}+h_{1}}{2}+h_{2}\right)(N_{6})\right]$$

### 2.4. Equations of Motion of the Sandwich Plate Element

#### 2.4.1. Potential Energy

The potential energy of the *i*th layer of the element due to stretching and bending are given by
(12)$$U_{i}=\frac{1}{2}\Delta^{e}{}^{{}^{T}}\left(h_{i}\int_{-a}^{a}\int_{-b}^{b}(B_{ei}{}^{T}D_{ei}B_{ei}+B_{bi}{}^{T}D_{bi}B_{bi})dxdy\right)\Delta^{e}=\frac{1}{2}\Delta^{e}{}^{{}^{T}}\left(K_{{}_{ei}}^{e}+K_{{}_{bi}}^{e}\right)\Delta^{e}$$
where the subscript i (i=1, 2, 3) indicates that the parameter belongs to the base plate, the viscoelastic layer and the constraint layer, respectively, $B_{ei}$ and $B_{bi}$ are the stretching and bending strain-displacement matrix, respectively. Their expressions are as follows:(13)$$\begin{array}{l} B_{e1}={\left[\begin{array}{ccc} \frac{\partial N_{3}}{\partial x} & \frac{\partial N_{4}}{\partial y} & \frac{\partial N_{3}}{\partial y}+\frac{\partial N_{4}}{\partial x} \end{array}\right]}^{T},\;B_{e2}={\left[\begin{array}{ccc} \frac{\partial N_{8}}{\partial x} & \frac{\partial N_{9}}{\partial y} & \frac{\partial N_{8}}{\partial y}+\frac{\partial N_{9}}{\partial x} \end{array}\right]}^{T},\\ B_{e3}={\left[\begin{array}{ccc} \frac{\partial N_{1}}{\partial x} & \frac{\partial N_{2}}{\partial y} & \frac{\partial N_{1}}{\partial y}+\frac{\partial N_{2}}{\partial x} \end{array}\right]}^{T},\;B_{bi}={\left[\begin{array}{ccc} \frac{\partial^{2}N_{5}}{\partial x^{2}} & \frac{\partial^{2}N_{5}}{\partial y^{2}} & 2\frac{\partial^{2}N_{5}}{\partial xy} \end{array}\right]}^{T} \end{array}$$

$D_{ei}$ and $D_{bi}$ are thein-plane stiffness matrices andthe bending stiffness matrices of the *i*th layer, respectively. They are given by
(14)$$D_{ei}=\frac{E_{i}}{1-\nu_{i}{}^{2}}\left[\begin{array}{ccc} 1 & \nu_{i} & 0\\ \nu_{i} & 1 & 0\\ 0 & 0 & \frac{1-v_{i}}{2} \end{array}\right],\;D_{bi}=\frac{E_{i}h_{i}{}^{3}}{12(1-\nu_{i}{}^{2})}\left[\begin{array}{ccc} 1 & \nu_{i} & 0\\ \nu_{i} & 1 & 0\\ 0 & 0 & \frac{1-\nu_{i}}{2} \end{array}\right]$$
where $E_{i}$ and $\nu_{i}$ are the elasticity modulus and Poisson’s ratio of the *i*th layer, respectively, $K_{ei}^{e}$ and $K_{bi}^{e}$ are the in-plane stretching and bending stiffness matrices of the *i*th layer, respectively. They are defined as follows:(15)$$K_{ei}^{e}=h_{i}\int_{-a}^{a}\int_{-b}^{b}(B_{ei}^{T}D_{ei}B_{ei})dxdy,\;K_{bi}^{e}=h_{i}\int_{-a}^{a}\int_{-b}^{b}(B_{bi}^{T}D_{bi}B_{bi})dxdy$$

The strain energy corresponding to shear of the viscoelastic layer can be written as
(16)$$U_{sv}=\frac{1}{2}\Delta^{e}{}^{{}^{T}}\left(h_{2}\int_{-a}^{a}\int_{-b}^{b}B_{sv}^{T}GB_{sv}dxdy\right)\Delta^{e}=\frac{1}{2}\Delta^{e}{}^{{}^{T}}K_{sv}^{e}\Delta^{e}$$
where $K_{sv}^{e}$ is the shear stiffness matrix of the viscoelastic layer, which is defined as
(17)$$K_{sv}^{e}=h_{2}\int_{-a}^{a}\int_{-b}^{b}B_{sv}^{T}GB_{sv}dxdy=G_{v}K_{v}^{e}$$
where $K_{v}^{e}$ is the viscous stiffness matrix, which is given by
(18)$$K_{v}^{e}=h_{2}\int_{-a}^{a}\int_{-b}^{b}(N_{10}{}^{T}N_{10}+N_{11}{}^{T}N_{11})dxdy$$

$B_{sv}$ is the shear strain-displacement matrix given by
(19)$$B_{sv}={\left[\begin{array}{cc} N_{10} & N_{11} \end{array}\right]}^{T}$$

G is the shear modulus matrix of the viscoelastic layer, which is defined as
(20)$$G=\left[\begin{array}{cc} G_{v} & 0\\ 0 & G_{v} \end{array}\right]$$
where $G_{v}$ is the shear modulus of the viscoelastic materials, which is generally in the form of a complex variable and is frequency-dependent.

Then, the total stiffness matrix $K^{e}$ is the sum of the stiffness matrices of each layer
(21)$$K^{e}=\underset{K_{e}^{e}}{\underbrace{\sum_{i=1}^{3}(K_{ei}^{e}+K_{bi}^{e})}}+\underset{K_{sv}^{e}}{\underbrace{G_{v}K_{v}^{e}}}$$
where $k_{e}^{e}=\sum_{i=1}^{3}(k_{ei}+k_{bi})$ is the elastic stiffness matrix of the element.

Obviously, the total potential energy of the element is the sum of the potential energy of each layer
(22)$$U=\sum_{i=1}^{3}U_{i}+U_{sv}$$

#### 2.4.2. Kinetic Energy

The kinetic energy of the *i*th (i=1, 2, 3) layer of the element due to stretching and bending are given by
(23)$$T_{i}=\frac{1}{2}\rho_{i}{∭}_{V}\left[{\left(\frac{\partial u_{xi}}{\partial t}\right)}^{2}+{\left(\frac{\partial u_{yi}}{\partial t}\right)}^{2}+{\left(\frac{\partial w_{i}}{\partial t}\right)}^{2}\right]dV$$
where $\rho_{i}$ is the density of the *i*th layer. At the right side of the equation, the sum of the first two terms is the tensile kinetic energy, and the third term is the bending kinetic energy.

The total kinetic energy of the element is the sum of the kinetic energy of each layer
(24)$$T=\sum_{i=1}^{3}T_{i}$$

Applying the shape functions, the total mass matrix $M^{e}$ of the element can be obtained as
(25)$$M^{e}=\sum_{i=1}^{3}\left(M_{ei}^{e}+M_{bi}^{e}\right)$$
where $M_{ei}^{e}$ and $M_{bi}^{e}$ are the stretching and bending mass matrix of the *i*th layer, respectively. The expressions for these mass matrices are given by
(26)$$\begin{array}{l} M_{e1}^{e}=\rho_{1}h_{1}\int_{-a}^{b}\int_{-a}^{b}\left(N_{3}{}^{T}N_{3}+N_{4}{}^{T}N_{4}\right)dxdy,\;M_{e2}^{e}=\rho_{2}h_{2}\int_{-a}^{b}\int_{-a}^{b}\left(N_{8}{}^{T}N_{8}+N_{9}{}^{T}N_{9}\right)dxdy\\ M_{e3}^{e}=\rho_{3}h_{3}\int_{-a}^{b}\int_{-a}^{b}\left(N_{1}{}^{T}N_{1}+N_{2}{}^{T}N_{2}\right)dxdy,\;M_{bi}^{e}=\rho_{i}h_{i}\int_{-a}^{b}\int_{-a}^{b}N_{5}{}^{T}N_{5}dxdy \end{array}$$

#### 2.4.3. Dynamic Equations of the Sandwich Plate Element

The equations of motion can be derived using Hamilton’s principle. The variational form of Hamilton’s principle can be expressed as
(27)$$\int_{t_{1}}^{t_{2}}\delta \left(T-U\right)dt+\int_{t_{1}}^{t_{2}}\delta Wdt=0$$
where $W=\Delta^{e}{}^{{}^{T}}F^{e}$ is the work done by the force of the element, where $F^{e}$ is the external force vector.

Substituting Equations (22) and (24) into Equation (27) gives
(28)$$M^{e}{\ddot{\Delta }}^{e}+K_{e}^{e}\Delta^{e}+G_{v}K_{v}^{e}\Delta^{e}=F^{e}$$

### 2.5. Convergence Analysis of the Element

In this section, an example will be used to analyze the convergence of the element. Consider a sandwich plate structure, whose boundary condition is fixed on the opposite side, as shown in Figure 4. Table 1 lists its material and geometric parameters.

The 28-degree-of-freedom element model here is used to calculate the natural frequencies and loss factors corresponding to the first three modes. When calculating, 3 × 3,4 × 4,5 × 5elements are divided along the length and width of the plate. The calculation results are shown in Table 2.

It can be seen from Table 2 that the element has good convergence. When the number of elements is nine, the calculation results of the system’s natural frequency and loss factor begin to converge obviously. When the number of elements reaches 16 and 25, the calculation results are basically unchanged. It can be considered that 16 elements completely meet the convergence requirements. This shows that the element has very good convergence characteristics.

### 2.6. Applying Biot Model

The Biot model can accurately describe the frequency-dependent characteristics of the viscoelastic materials. In Biot model, a series of mini-oscillator terms are used to describe the shear modulus function $G_{v}$ of the viscoelastic materials [29]. In the Laplace domain, its expression is [30]
(29)$$sG_{v}(s)=G^{\infty }\left[1+\sum_{i=1}^{n}\frac{a_{i}s}{s+b_{i}}\right]$$
where $G^{\infty }$ represents the equilibrium value of the shear modulus. *n* is the number of the mini-oscillator terms, $\left\{a_{i},b_{i}\right\}$ with i=1, 2, 3, …n as positive constants. These parameters can be determined by the following nonlinear curve fitting method.

In Equation (29), let,
S=jw, one gets
(30)$$G_{v}(j\omega )=G^{\infty }\left[1+\sum_{i=1}^{n}\frac{a_{i}\left(jw\right)}{\left(jw\right)+b_{i}}\right]$$

The nonlinear curve fitting function expression in the frequency domain can be written as
(31)$$F(x)=\sum_{i=1}^{m}{\left|G_{v}\left(x,\;\omega_{i}\right)-G_{0}\left(\;\omega_{i}\right)\right|}^{2}=\min$$
where $G_{v}\left(x,\;\omega_{i}\right)$ is the Biot model with parameters to be determined, $G_{0}\left(\;\omega_{i}\right)$ is the measured complex modulus value in the complex frequency domain or other viscoelastic material damping model expressions obtained from experimental data, m is the number of the measured complex modulus value, x with $x_{i}>0,\;i=1,\;2,\;3\cdot \cdot \cdot 2n+1$ is the parameters of the Biot model to be determined, and its expression is
(32)$$x_{1}=G^{\infty };\;x_{2}=a_{1},\;x_{3}=a_{2},\;\cdot \cdot \cdot x_{n+1}=a_{n};\;x_{n+2}=b_{1},\;x_{n+3}=b_{2},\;\cdot \cdot \cdot x_{2n+1}=b_{n}$$

Solving the above optimization problems, one can obtain the Biot model parameters of viscoelastic materials.

Carrying on Laplace transform to Equation (28), one gets
(33)$$\left(s^{2}M^{e}+K_{e}^{e}+sG_{v}\left(s\right)K_{v}^{e}\right)\Delta^{e}\left(s\right)=F^{e}\left(s\right)$$

Substituting Equation (29) into Equation (33) and introducing the auxiliary dissipation coordinates
(34)$${\hat{Z}}_{i}\left(s\right)=\frac{b_{i}}{s+b_{i}}\Delta^{e}\left(s\right),\;i=1,\;2,\;3\cdot \cdot \cdot N$$
one can obtain the equation of motion of the sandwich plate elements incorporating the Biot model
(35)$$\tilde{M}\ddot{q}+\tilde{C}\dot{q}+\tilde{K}q=\tilde{F}$$
where
(36)$$\begin{array}{c} \tilde{M}=\left[\begin{array}{cccc} M^{e} & 0 & \cdots & 0\\ 0 & 0 & \cdots & 0\\ \vdots & \vdots & \ddots & \vdots \\ 0 & 0 & \cdots & 0 \end{array}\right],\;\tilde{C}=\left[\begin{array}{cccc} 0 & 0 & \cdots & 0\\ 0 & \frac{a_{1}}{b_{1}}\Lambda & \cdots & 0\\ \vdots & \vdots & \ddots & \vdots \\ 0 & 0 & \cdots & \frac{a_{n}}{b_{n}}\Lambda \end{array}\right],\;\\ \tilde{K}=\left[\begin{array}{cccc} K_{e}^{e}+A\left(1+\sum_{k=1}^{n}a_{k}\right) & -a_{1}B & \cdots & -a_{n}B\\ -a_{1}B^{T} & a_{1}\Lambda & \cdots & 0\\ \vdots & \vdots & \ddots & \vdots \\ -a_{n}B^{T} & 0 & \cdots & a_{n}\Lambda \end{array}\right],\;q=\left\{\begin{array}{c} \Delta^{e}\\ Z_{1}\\ \vdots \\ Z_{n} \end{array}\right\},\;\tilde{F}=\left\{\begin{array}{c} F^{e}\\ 0\\ \vdots \\ 0 \end{array}\right\} \end{array}$$
where $A=G^{\infty }k_{v}^{v}$, $k_{v}^{e}=B_{v}\Lambda_{v}B_{v}^{T}$, $\Lambda =G^{\infty }\Lambda_{v}$, $B=B_{v}\Lambda$, $Z_{j}=B_{v}^{T}{\hat{Z}}_{j},\;(j=1,\;2,\;\cdots ,n)$, where $\Lambda_{v}$ is a diagonal matrix composed of the positive eigenvalues of the viscosity stiffness matrix $k_{v}^{v}$, $B_{v}$ is the matrix with corresponding orthogonal eigenvectors as columns.

Equation (35) is the dynamic equation incorporating the Biot model of the sandwich plate element. According to the general element integration method in the finite element theory, integrating the physical coordinates **X** of the sandwich plate structure, one can obtain the overall dynamic equation of the sandwich plate structure as follows
(37)$$M\ddot{X}+C\dot{X}+KX=F$$
where M,  C and K are the total mass, damping and stiffness matrices of the sandwich plate, and F is the excitation force.

Obviously, Equation (37) is a general second-order, steady-state linear system dynamics equation. It is very convenient to solve the natural frequency, damping and other modal parameters, which makes the Biot model have good engineering application value.

### 2.7. Non-Linear Eigenvalue Problem

After applying the Biot model, the structural dynamics equation contains the physical nonlinear of the viscoelastic material, and the vibration is nonlinear, the eigenvalue problem is nonlinear as well. Therefore, in order to solve this problem, Equation (37) needs to be decoupled and transformed from the second-order differential equation to the first-order state equation.

Set $y={\left\{X\;\dot{X}\right\}}^{T}$,introduce auxiliary equation $M\dot{X}-M\dot{X}=0$, Equation (37) can be written as
(38)$$\bar{A}\dot{y}+\bar{B}y=\bar{F}$$
where
(39)$$\bar{A}=\left[\begin{array}{cc} C & M\\ M & 0 \end{array}\right],\;\bar{B}=\left[\begin{array}{cc} K & 0\\ 0 & -M \end{array}\right],\;\bar{F}=\left[\begin{array}{c} F\\ 0 \end{array}\right]$$

In the free vibration, F=0, and the Equation (38) can be expressed as
(40)$$\bar{A}\dot{y}+\bar{B}y=0$$

With the mathematical software MATLAB, it is easy to solve the eigenvalue problem of Equation (40) to get the complex eigenvalue matrix:(41)$$\left[\begin{array}{ccc} \ddots & & \\ & \lambda & \\ & & \ddots \end{array}\right]=\left[\begin{array}{ccccc} \lambda_{1} & & & & \\ & \lambda_{1}^{*} & & & \\ & & \ddots & & \\ & & & \lambda_{Ν} & \\ & & & & \lambda_{Ν}^{*} \end{array}\right]$$
Then the natural frequency and loss factor are determined by [31]
(42)ωN=Im(λN)2−Re(λN)2, ηN=−2Re(λN)Im(λN)Im(λN)2−Re(λN)2

## 3. Numerical Simulation and Validation

Three cantilever sandwich plate structures with different lengths are considered here. Their geometry and parameters are shown in Table 3. In reference [32], a series of experiments were carried out on them to determine the mechanical properties of viscoelastic materials and the vibration properties of the plates. In this section, the finite element method developed is used to analyze these plates, and the first three natural frequencies and loss factors are obtained. The results are compared with the experimental values to verify the finite element model in this paper.

In Table 3, the elastic modulus of the viscoelastic materials are frequency-dependent and the reference [32] experimentally determined their expressions as
(43)$$E*(\omega )=\varepsilon +\frac{\alpha \omega^{2}\left(-\omega^{2}+i\beta \omega \right)}{-\omega^{2}+i\beta \omega +\delta }$$
where α=5.26MPa, $\beta =55.59\times 10^{6}s^{-1}$, $\delta =6.98\times 10^{9}s^{-2}$, ε=0.58MPa.

According to Equation (43), one can get the measured complex modulus value $G_{0}\left(\;\omega_{i}\right)$ of the viscoelastic materials, and then substituting them into Equation (31), by nonlinear curve fitting, and the parameters of the Biot model can be obtained as shown in Table 4. Figure 5, Figure 6 and Figure 7 show the comparison of the fitted Biot model and the experimental value in reference [32]. The fitting of the real and imaginary parts are shown in Figure 5 and Figure 6, respectively. The fitting error is shown in Figure 7.

It can be seen from Figure 5, Figure 6 and Figure 7 that, when three micro-vibrators are used, the Biot model can well simulate the modulus of viscoelastic materials. The fitting accuracy of real and imaginary parts is very good. In a wide frequency band of 10 to 500 Hz, the fitting errors of the imaginary and real parts are below 3%. This proves that the fitting method presented here is correct.

The finite element method presented this paper is used to calculate the first three order natural frequencies and loss factors of the three sandwich plates with different lengths, respectively, and the results are listed in Table 5. When calculating, all the plates are divided into 20×5 elements along the length and width directions. All solving processes are completed by self-programming with MATLAB software.

It can be seen from Table 5 that, when the lengths of the sandwich plates are increased from 500to 1000mm, the natural frequencies and loss factors of the plate system are significantly changed, which indicates that the vibration characteristics of the sandwich plate structure are very sensitive to the length.

In addition, as can be seen from Table 5, in the estimation of the natural frequencies, the accuracy of the finite element model in this paper for all three plates is less than 3%, the minimum error is 1.32%, the maximum error is 2.34%, and the average error is 1.88%. In the structural loss factor estimation, the prediction accuracy of the model is below 4%, the minimum error is 2.10%, the maximum error is 3.40%, and the average error is 2.62%. Therefore, the method in this paper can be considered to be accurate and effective.

In addition, the reference [32] also provides a GHM-based sandwich finite element model to obtain the first three natural frequencies of the three sandwich structures. In order to further verify the finite element of this paper, the calculation results of the two numerical methods are listed in Table 6.

It can be seen from Table 6 that the differences between these results are inferior to 15.60%. Such differences may be explained by the quality of the curve fitting. The GHM curve fitting of the reference [32] is four parameters, and the Biot model curve fitting presented this paper is seven parameters. In general, in terms of calculation accuracy, the results of the finite element method of this paper better agree with the experimental values than the GHM-based sandwich finite element model of the reference [32].

## 4. Experimental Validation and Comparison

A clamped-free sandwich plate is used to validate the accuracy of the FE model. Figure 8 shows the experimental set-up. The sandwich plate is 280 mm long and 200mm wide. The constraining layer is an aluminum sheet. The base plate is 45 steel. The core layer is ZN-1 viscoelastic materials made in the Chinese Aerospace Research Institute of Materials & Processing Technology. The physical and geometrical parameters of the sandwich plate are shown in Table 7.

The sandwich plate is excited by an impact hammer with a hard hammerhead. The top displacement signal of the plate is measured by a laser displacement sensor (LK-G500, Keyence Corporation, Osaka, Japan) at the middle-end of the test plate which has an accuracy of 0.005 µm over a frequency band between 0 and 392 KHz. The laser displacement sensor is equipped with data acquisition software named LK-Navigator which is installed on the computer. Through the acquisition software, one can issue commands to the controller to control the sampling points and the sampling frequency. In the experiment, the sampling points are set to 50,000 and the sampling frequency is set to 5000Hz.

During the experiment, first fix the laser displacement sensor on a liftable bracket, then connect the laser displacement sensor and its controller (the controller is connected to the DC 24V power supply), and the controller is connected to the computer. After the power is turned on, adjust the distance between the laser displacement sensor and the sandwich plate and the horizontal position of the sensor, and ensure that the distance of the laser displacement sensor and the sandwich plate is within the specified range, so that the indicator light of the sensor lights normally, and the controller screen displays green numbers.

Open the measurement software in the computer, and then hit the different positions of the sandwich plate with the hammer. The software will automatically collect the displacement and time data measured by the laser displacement sensor. Repeat five times at each position and average the measured data to consider the variability in the measurements. Finally, the Matlab software is used to process the experimental data, the natural frequency is obtained by Fourier transform, and the loss factor value is obtained by the half power method.

The theoretical modal parameters, the first two natural frequencies and associated loss factors, evaluated by the finite element method, are experimentally verified at 30 °C. In the theoretical calculations, the Biot model parameters of the ZN-1 viscoelastic material in Table 7 are taken from reference [33]. The sandwich plate is divided into 10×8elements. The natural frequencies and associated loss factors of the first two modes are presented in Table 8.

The results presented in Table 8 show a good agreement between the theoretical predictions and experimental values for the first two natural frequencies and loss factors. The average error of the natural frequencies is 2.25%, and all errors are below 3%. The average error of the loss factors is 4.75%, and all errors are below 5%. The major reasons for the error are twofold: first, it is difficult to achieve strict clamping in the experiment; second, some simplified assumptions are made in the modeling process. However, in general, the errors are within the acceptable range, which can prove that the finite element method presented is correct and effective.

## 5. Conclusions

A finite element modeling is developed for the sandwich plate with a frequency-dependent viscoelastic material core. The elements are three-layer composite quadrilateral plate elements bounded by four nodes with seven degrees of freedom per node. The Biot model is used to describe the frequency dependent properties of the viscoelastic materials. The method to determine the parameters of the Biot model is presented. The Biot model is combined into the finite element dynamic equation of the sandwich plate structure by introducing auxiliary coordinates. The new dynamic equation is transformed into a second-order linear system form by matrix transformation, which reduces the difficulty of solving the conventional nonlinear system equation. The finite element method is verified by numerical simulation and experimental research. The results show that the finite element model has a good accuracy in predicting the natural frequencies and loss factors of the sandwich plate structures.

At present, this method is limited to the viscoelastic sandwich plate structure with isotropic material layers. In the next step, the method will be improved to make it suitable for a viscoelastic sandwich plate with anisotropic materials.

## Figures and Tables

**Figure 1 materials-13-02296-f001:**
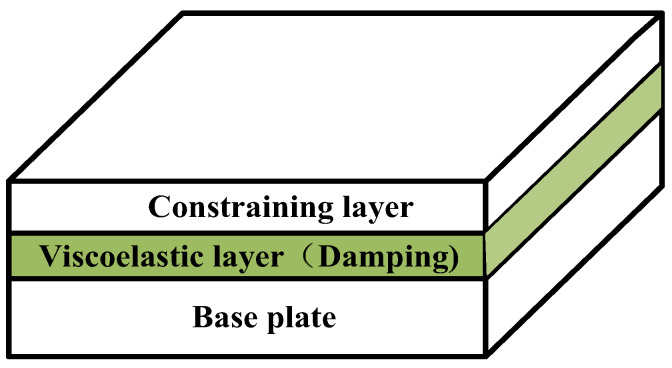
Viscoelastic sandwich plate structure.

**Figure 2 materials-13-02296-f002:**
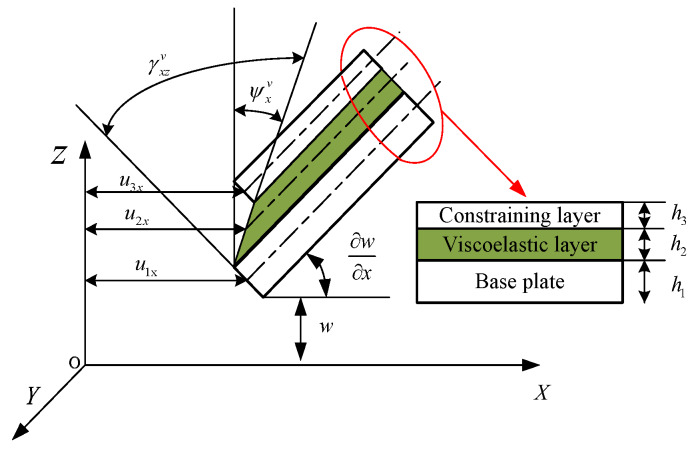
Geometry and deformation of sandwich plate.

**Figure 3 materials-13-02296-f003:**
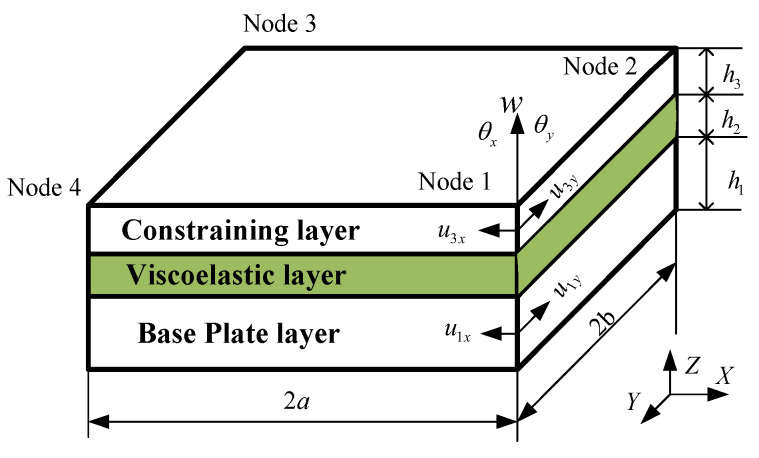
The element of sandwich plate.

**Figure 4 materials-13-02296-f004:**
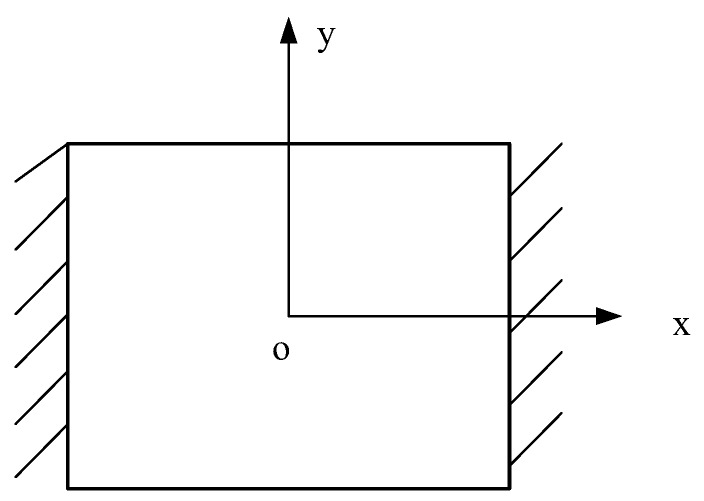
The boundary condition of the sandwich plate.

**Figure 5 materials-13-02296-f005:**
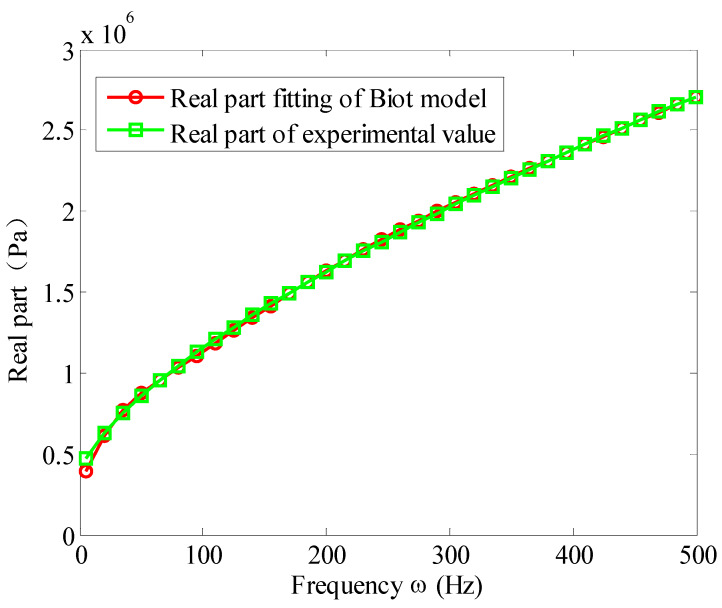
The fitting of the real part.

**Figure 6 materials-13-02296-f006:**
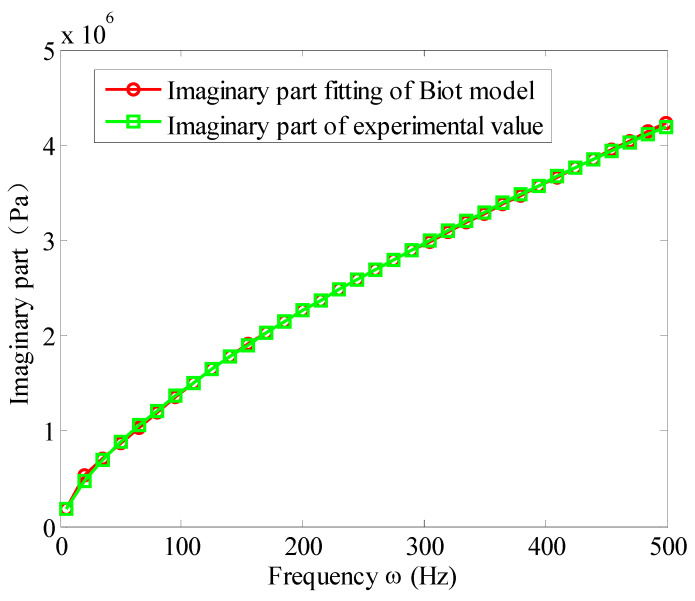
The fitting of the real part.

**Figure 7 materials-13-02296-f007:**
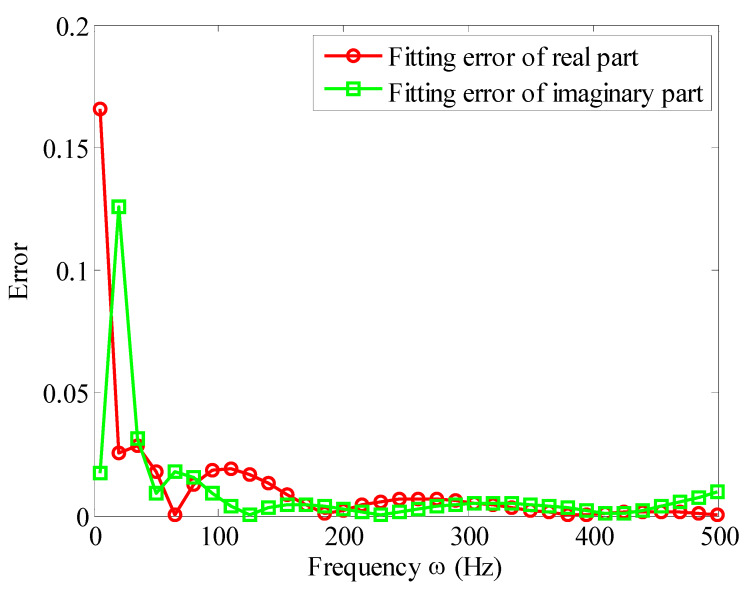
The fitting errors of the real and imaginary parts.

**Figure 8 materials-13-02296-f008:**
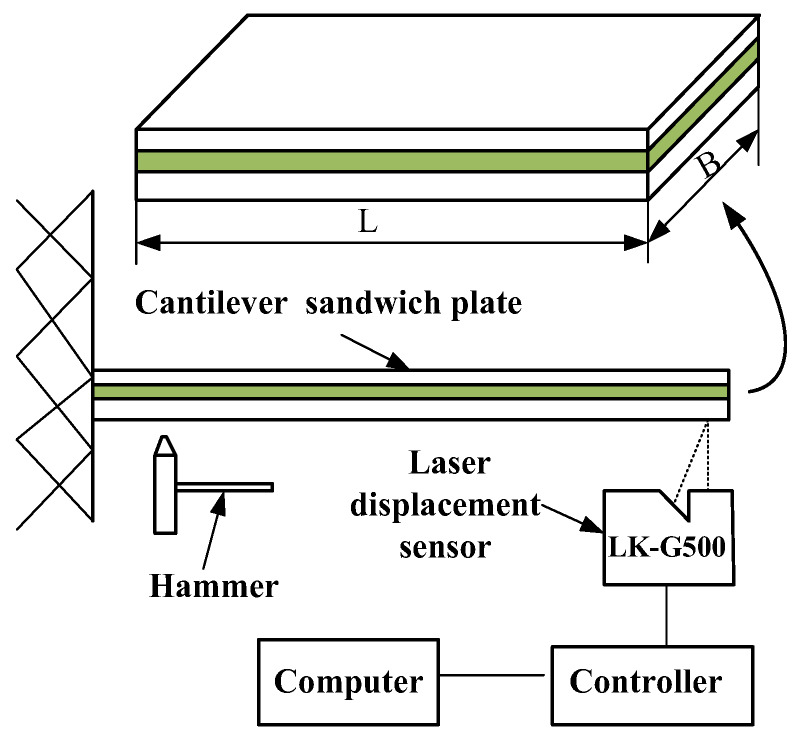
The schematic diagram of the experimental set-up.

**Table 1 materials-13-02296-t001:** Material and geometric parameters of the sandwich plate.

Material Properties	Constraining Layer	Base Plate	Viscoelastic Core
Elastic Modulus(GPa)	71	71	0.000896
Density(kg/m^3^)	2700	2700	999
Poisson’s ratio	0.3	0.3	0.498
Loss factor	-	-	0.9683
Thickness (mm)	1.0	3.0	1.0
Length (mm)	400	400	400
Width (mm)	400	400	400

**Table 2 materials-13-02296-t002:** The frequencies and loss factors versus the number of elements.

Modes	9 Element (3 × 3)		16Element (4 × 4)		25Element (5 × 5)	
	Natural Frequencyω(Hz)	Loss Factorsη	Natural Frequencyω(Hz)	Loss Factorsη	Natural Frequencyω(Hz)	Loss Factorsη
1	95.96	0.1331	95.09	0.1315	94.97	0.1295
2	112.93	0.1303	112.70	0.1274	112.70	0.1270
3	187.62	0.1431	187.25	0.1397	187.24	0.1402

**Table 3 materials-13-02296-t003:** Geometric and material parameters of cantilever viscoelastic sandwich plate.

Material Properties	Constraining Layer	Base Plate	Viscoelastic Layer
Young’s modulus(GPa)	68.7	68.7	Frequency-dependent
Density(kg/m^3^)	2690	2690	795
Poisson’s ratio	0.3	0.3	0.3
Thickness (mm)	3	3	2
Length (mm)	*L*_1_ = 500, *L*_2_ = 800, *L*_3_ = 1000		
Width (mm)	24	24	24

**Table 4 materials-13-02296-t004:** The Biot model parameters of viscoelastic materials.

k	$G^{\infty }$	$a_{k}$	$b_{k}$
1	$5.8\times 10^{5}$	2.8378	151.9889
2		0.0552	151.9900
3		2.6365	2.3122e6

**Table 5 materials-13-02296-t005:** The first three natural frequencies and loss factors of viscoelastic sandwich plates of different lengths: comparison of experimental values and numerical simulation results.

Length	Modal Order	Experimental Results [32]		Results of the Finite Element Method of This Paper			
		Natural Frequency (Hz)	Loss Factor	Natural Frequency (Hz)	Error (%)	Loss Factor	Error (%)
*L*_1_=500mm	1	16.95	0.1748	17.20	1.77	0.1790	2.40
	2	79.33	0.1350	80.51	1.48	0.1382	2.37
	3	184.44	0.0765	187.80	1.82	0.0791	3.40
*L*_2_=800mm	1	7.55	0.1770	7.65	1.32	0.1818	2.71
	2	37.13	0.1768	38.00	2.34	0.1807	2.21
	3	93.18	0.0788	95.52	2.51	0.0808	2.54
*L*_3_=1000mm	1	5.05	0.1434	5.12	1.38	0.1431	2.10
	2	24.54	0.1508	25.03	2.00	0.1540	2.12
	3	60.13	0.1754	61.54	2.34	0.1819	3.71

**Table 6 materials-13-02296-t006:** Comparison of the numerical simulation results of the GHM-based sandwich finite element model and the finite element method of this paper.

Length	Modal Order	Results of the GHM-Based Sandwich Finite Element Model [32]	Results of the Finite Element Method of This Paper	
		Natural Frequency (Hz)	Natural Frequency (Hz)	Error (%)
*L*_1_=500mm	1	14.86	17.20	13.6
	2	88.75	80.51	10.2
	3	166.83	187.80	11.2
*L*_2_=800mm	1	6.61	7.65	13.4
	2	32.05	38.00	15.6
	3	83.16	95.52	12.4
*L*_3_=1000mm	1	4.32	5.12	15.6
	2	22.88	25.03	8.6
	3	53.66	61.54	12.8

**Table 7 materials-13-02296-t007:** Physical and geometrical properties of the base plate, the viscoelastic layer and the constraining layer.

Material Properties	Constraining Layer	Base Plate	Viscoelastic Layer
Young’s modulus(GPa)	71	210	Frequency-dependent
Density(kg/m^3^)	2710	7850	1000
Poisson’s ratio	0.3	0.3	0.3
Thickness (mm)	1	1	2

**Table 8 materials-13-02296-t008:** Natural frequencies and loss factors of the sandwich plate.

Mode Order	Experimental		Finite Element Model			
	Natural Frequencies *ω*(Hz)	Loss Factors *η*	Natural Frequencies *ω*(Hz)	Error (%)	Loss Factors *η*	Error (%)
1	24.73	0.218	25.21	1.9	0.228	4.6
2	69.17	0.098	71.00	2.6	0.102	4.1

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
