# Peer review of "TheFinite Element Modeling and Experimental Study of Sandwich Plates with Frequency-Dependent Viscoelastic Material Model"

_materials, 2020, doi:10.3390/ma13102296_

Round 1

Reviewer 1 Report

The article deals with finite element modeling for three-layered frequency-dependent viscoelastic plates and comparison with experimental data. Some issues have to be fixed before the article can be published in Materials.

General comments

  • The English of the article is very poor and must be improved. There are many typos and grammatical mistakes turning the text difficult to understand.
  • The authors should state clearly the novelties of their article compared with previous works.
  • The article is specific to the Biot viscoelastic model, the title should be changed accordingly.

Specific comments

Introdution

The bibliography on frequency-dependent viscoelastic sandwich plates finite element modeling, parameter identification, and experimental vibration analysis is not fully covered and many references have to be added. For example, the authors should consider adding the following references :

  • Ayodele Adessina, Mohamed Hamdaoui, Chao Xu, El Mostafa Daya, « Damping properties of bi-dimensional sandwich structures with multi-layered frequency-dependent visco-elastic cores », Composite Structures, Volume 154, 2016, Pages 334-343.
  • Hamdaoui, K.S. Ledi, G. Robin, E.M. Daya, Identification of frequency-dependent viscoelastic damped structures using an adjoint method, Journal of Sound and Vibration, Volume 453, 2019, Pages 237-252.
  • Mohamed Hamdaoui, Komlan Akoussan, El Mostafa Daya, Comparison of non-linear eigensolvers for modal analysis of frequency dependent laminated visco-elastic sandwich plates,Finite Elements in Analysis and Design,Volume 121,2016,Pages 75-85.
  • Wang, T., Xu, C., Hamdaoui, M., Guo, N., & Gu, L. (2019). Uncertainty propagation in modal analysis of viscoelastic sandwich structures using a stochastic collocation method. Journal of Sandwich Structures & Materials.
  • Komlan Akoussan, Mohamed Hamdaoui, El Mostafa Daya,Improved layer-wise optimization algorithm for the design of viscoelastic composite structures, Composite Structures,Volume 176,2017,Pages 342-358.

Finite element modeling

  • The authors should mention which kinematics hypothesis they used to form the model. Is it Kirchoff-Love or Mindlin-Reissner or zig-zag kinematics?
  • Convergence analysis for frequencies and damping ratios versus the number of elements should be provided to assess the numerical validation of the model.
  • The author should mention the numerical method used to solve for the frequencies and damping ratios.

Numerical simulation and validation

  • When fitting the Biot model to the experimental GHM model of reference [23], the authors should provide some convergence diagnostics (residual errors, etc.)
  • The authors should also mention the numerical method used, the implementation software, etc.
  • The authors compare their finite element model to the experimental data in [23] for the three first modes. Why not use more modes?

Experimental validation and comparison

  • The authors should be more specific about the experimental method and especially the technique they used to obtain frequencies and damping ratios from experimental data
  • Here the authors compare their finite element model and their experimental data for only two modes. Why is it so? Mre modes could be used ?

Reviewer 2 Report

This manuscript is a description of combining experimental measurement with finite element methods and considering vibration by Biot's model. Unfortunately I cannot find any reasons to publish this article, major part of the text was consider basics of the field, measurements was quite simple and conclusions says that it is possible to use FEM in this kind of problems, which is obvious. Moreover text was partly weakly written and some equations were broken (there are some technical issues related to typing). There was not a single figure of the simulations. Results was presented in tables but there was also extra numbers to confuse reader. In overall, I consider this paper weak, novelty is very questionable and therefore reject it. 

Reviewer 3 Report

In this article, Huang et al have studied the effects of viscoelastic material core on vibrational analysis of sandwich plate. The study is interesting. However, authors have not done the literature survey properly. I suggest authors to include these relevant articles related to the viscoelastic properties of materials in their literature (ACS Nano 2018, 12, 7, 6378-6388; ACS Nano 2017, 11, 5, 5148-5159; Angew.Chem.Int. Ed. 2019 , 58,18562 –18569). I would also suggest to look for other relevant articles to include in the reference list, before it is considered as suitable for publication. 

Round 2

Reviewer 1 Report

The authors have made substantial changes to their manuscript.

However, the following points must be addressed before publication :

1- The English need some improvements. I advise reviewing the article by an English native speaker.

2- In the added paragraph on nonlinear eigenvalue problems, there are several mistakes. The auxiliary equation (line 282) should be corrected. Equations (40), (41), (43) and (44) are not correct. The authors should provide references for these expressions or explain their approach.

3- Line 316, there is a typo for the Biot model

4- The authors should include a comparison of the predictions of their numerical model with previous results in the literature for three-layered viscoelastic sandwich structures. The can rely on ref [21] for example.

5- The authors should add perspectives to their work based on the limitations of their approach. The variability in the measurements should be considered.

Reviewer 2 Report

Unfortunately the changes that authors have done, does not change my opinion related to this paper. Novelty of the paper is questionable and measurements are pretty simple. Adding extra references does not help in this case. 
